# Inhibition of Bacterial Adhesion and Biofilm Formation by Seed-Derived Ethanol Extracts from *Persea americana* Mill

**DOI:** 10.3390/molecules27155009

**Published:** 2022-08-06

**Authors:** Silvia del Carmen Molina Bertrán, Lianet Monzote, Davie Cappoen, Julio Cesar Escalona Arranz, Mario Juan Gordillo Pérez, Annarli O. Rodríguez-Ferreiro, Idelsy Chill Nuñez, Claudina Pérez Novo, Daniel Méndez, Paul Cos, Gabriel Llauradó Maury

**Affiliations:** 1Pharmacy Department, Faculty of Natural and Exact Sciences, University of Oriente, Santiago de Cuba 90500, Cuba; 2Department of Parasitology, Institute of Tropical Medicine “Pedro Kourí”, Havana 11400, Cuba; 3Laboratory of Microbiology, Parasitology and Hygiene (LMPH), Faculty of Pharmaceutical, Biomedical and Veterinary Sciences, University of Antwerp, 2610 Antwerp, Belgium; 4Biology Department, Faculty of Natural and Exact Sciences, University of Oriente, Santiago de Cuba 90500, Cuba; 5Department of Biomedical Engineering, Faculty of Telecom, Informatics and Biomedical Engineering, University of Oriente, Santiago de Cuba 90500, Cuba; 6Laboratory for Protein Chemistry, Proteomics and Epigenetic Signaling, Faculty of Pharmaceutical, Biomedical and Veterinary Sciences, University of Antwerp, 2610 Antwerp, Belgium; 7Chemistry Department, Faculty of Applied Sciences, University of Camagüey, Carretera de Circunvalación Km 512, Camagüey 74650, Cuba; 8Center of Studies for Industrial Biotechnology (CEBI), University of Oriente, Santiago de Cuba 90500, Cuba

**Keywords:** bacterial adhesion, biofilm, virulence factors, avocado seed, ethanol extracts

## Abstract

The increase in antibiotic resistance demands innovative strategies to combat microorganisms. The current study evaluated the antibacterial and antivirulence effects of ethanol extracts from *Persea americana* seeds obtained by the Soxhlet (SE) and maceration (MaE) methods. The UHPLC-DAD-QTOF analysis showed mainly the presence of polyphenols and neolignan. Ethanol extracts were not cytotoxic to mammalian cells (CC_50_ > 500 µg/mL) and displayed a moderate antibacterial activity against *Pseudomonas aeruginosa* (IC_50_ = 87 and 187 µg/mL) and *Staphylococcus aureus* (IC_50_ = 144 and 159 µg/mL). Interestingly, no antibacterial activity was found against *Escherichia coli.* SE and MaE extracts were also able to significantly reduce the bacterial adhesion to A549 lung epithelial cells. Additionally, both extracts inhibited the biofilm growth at 24 h and facilitated the release of internal cell components in *P. aeruginosa*, which might be associated with cell membrane destabilization. Real-time PCR and agarose electrophoresis gel analysis indicated that avocado seed ethanol extracts (64 µg/mL) downregulated virulence-related factors such as *mexT* and *lasA* genes. Our results support the potential of bioproducts from *P. americana* seeds as anti-adhesive and anti-biofilm agents.

## 1. Introduction

Infectious diseases continue to be an important health concern and cause of economic burden and low life expectancy in many developing countries [1,2]. Antimicrobial drugs have been developed and used for decades to beat pathogenic microbes, but the increase in antimicrobial resistance demands novel approaches [3,4].

The capacity of bacteria to adhere to cells enables them to colonize and invade tissues, thereby facilitating the biofilm formation and the establishment of infections [5,6]. Bacterial adhesion has been recognized as the first stage prior to biofilm development and a key step in pathogenesis. Therefore, adhesion becomes an important stage to eliminate the pathogens before a biofilm is organized and well-structured [7]. Pulmonary, vaginal, and urinary tract infections are some of the most common problems associated with bacterial attachment and biofilm formation [8,9]. In this sense, pharmacologically active substances that could interrupt bacterial adhesion, quorum sensing and biofilm architecture formation, constitute an important approach to develop new antibacterial agents [3,10].

Bioactive extracts and isolated molecules derived from natural sources have shown a huge potential to prevent biofilm formation by affecting the viability and adhesion of bacteria [11,12]. Several studies revealed that natural bioproducts, which target virulence factors, could modify some physicochemical and structural characteristics in bacteria such as surface charge and hydrophobicity, alterations on membrane components, and downregulating the adhesion molecules expression. This could help to disrupt cell aggregation, attachment, and the biofilm architecture. Some of the most clinically relevant microorganisms for their adhesion potential are *Pseudomonas aeruginosa*, *Escherichia coli*, and *Staphylococcus aureus* [13,14].

*Persea americana* Mill. (Lauraceae), commonly known as avocado, is a well-studied tropical tree from a nutritional and pharmacological point of view [15,16]. The avocado seeds have been traditionally used as remedies to treat asthma, inflammation, and mainly diarrhea and infectious diseases [16,17]. These effects have been mainly associated to the presence of secondary metabolites such as polyphenols [18]. In a previous study, we demonstrated that ethanol extracts from the seeds of *P. americana* growing in the Cuban eastern region showed a phytochemical composition rich in polyphenols such as catechins and neolignans derivatives as well as some fatty acids/alcohols as main [19]. However, in spite of its antimicrobial potential, avocado seed-derived bioproducts have been scarcely explored for their activity against bacterial adhesion and biofilm formation. Therefore, in this research we evaluated the antimicrobial potential of these two well characterized ethanol extracts from *P. americana* seeds against clinically relevant Gram-negative and Gram-positive bacteria by exploring their effect on bacteria viability, and their inhibitory activity on A549 lung epithelial cells adhesion and biofilm formation as well. Finally, the effect of the ethanol extracts on membrane disruption and mRNA expression of genes related to virulence and adhesion-associated structures in *P. aeruginosa* were also assessed.

## 2. Results

### 2.1. Plant Samples Preparation

The phytochemical analysis validated many phytoconstituents in varying concentrations. The two extracts: Soxhlet (SE) and Maceration/Stirring (MaE) showed similar phytochemical composition, based on their chromatographic similarity (Appendix A). However, the SE extract showed more peaks than MaE (see Appendix A). In general, the peak assignment was almost the same as in previous work [19], confirming that the chemical profile of these extracts does not change significantly between batches and that the standardized preparation procedure used is effective. Moreover, it was shown that the seeds of *P.americana* are rich in polyphenols such as catechins, lignans, and neolignan derivatives.

### 2.2. Antibacterial Activity and Cytotoxicity of P. americana Seed Ethanol Extracts

The in vitro antibacterial activity of *P. americana* seed extracts against reference bacteria strains was assessed by microdilution bioassays (Table 1). Both extracts showed IC_50_ values lower than 200 µg/mL for *P. aeruginosa* ATCC 9027 and *S. aureus* ATCC 6538. No antibacterial activity was found against *E. coli* ATCC 8739. For *S. pneumoniae* NTCC 7466, the SE showed an IC_50_ value of 227.6 µg/mL, while the MaE was not active with an IC_50_ value higher than 500 µg/mL. In general, SE showed IC_50_ values lower than MaE. Moreover, no significant cytotoxicity on mammalian cells was found (Table 2).

### 2.3. Ethanol Extracts from P. americana Seeds Inhibit Bacterial Adhesion to A549 Epithelial Lung Cells

To investigate the adherence/anti-adherence properties of bacteria co-culture assays are a valuable approach [20]. The effect of *P. americana* seed ethanol extracts to interrupt bacterial adhesion to A549 lung epithelial cells was assessed after 60 min of incubation (Figure 1). The concentrations used to explore this activity were fixed one dilution lower than their IC_50_ values obtained in the antibacterial screening. In general, ethanol extracts showed activity against both Gram-negative and Gram-positive bacteria. At the concentrations tested, all strains were reduced in their capacity to adhere A549 epithelial lung cells. The extracts statistically (*p* < 0.05) inhibited the *P. aeruginosa* adherence at 64 µg/mL, confirming the susceptibility of this strain to the ethanol extracts. Particularly, SE displayed a higher activity compared to MaE. For *S. aureus* adherence, SE and MaE were also noteworthy active. Although for *S. pneumoniae* and *E. coli* higher levels of concentrations were explored (128 and 512 µg/mL, respectively), the extracts inhibited the bacteria adherence. Nonetheless, SE was statistically more active against *S. pneumoniae* than MaE at dose 128 µg/mL.

### 2.4. Ethanol Extracts from P. americana Seeds Inhibits P. aeruginosa Biofilms

Biofilms are a dominant microbial lifestyle implicated in treatment failure for several infectious diseases. Therefore, studies to discover natural anti-biofilm agents constitute an interesting approach [21]. Since the highest antibacterial activity was found against *P. aeruginosa*, the *P. americana* seed ethanol extracts were evaluated for their activity against *P. aeruginosa* biofilms (Figure 2). Both ethanol extracts significantly inhibited the initial stage of biofilm formation (*p* < 0.05) in comparison with untreated bacteria (control) even at very low concentrations up to 8 µg/mL.

### 2.5. Ethanol Extracts from P. americana Seeds Affect P. aeruginosa Membrane Integrity

The potential damage on membrane integrity of *P. aeruginosa* treated with SE and MaE was assessed by measuring the release of 260 nm absorbing materials, including nucleic acids, at 6 and 24 h of treatment (Figure 3). After 6 h, low levels of DNA detected indicate that the membrane integrity is partially affected. A significant nuclei acid leakage was shown (*p* < 0.05) when bacteria were exposed to the extracts for 24 h. These results suggest a disruption of the bacterial membrane after treatment with the extracts. The activity observed in this study at 24 h is in concordance with the results obtained on *P. aeruginosa* (IC_50_) in the direct antibacterial assay.

### 2.6. Effect of Ethanol Extracts from P. americana Seeds on Virulence and Adhesion-Related Genes of P. aeruginosa

The effect of SE and MaE extracts on some pathogenic features of *P. aeruginosa* was examined by the expression of virulence and adhesion-related genes. In total, 10 virulence-related genes were studied and analyzed by RT-PCR [22] and the results were expressed by relative mRNA expression levels of genes calculated through the value obtained between Ct values of treated and untreated bacteria (Control) (Figure 4). The study showed that SE strongly downregulated the *lasA* and *mexT* genes expression (*p* < 0.05), compared with the control. On the contrary, MaE suppressed only the *mexT* gene expression. These effects were confirmed by the gel electrophoresis analysis, where the presence of *lasA* and *mexT* genes bands were not detected (Figure 5). On the other hand, no inhibitory effect was found for the other genes.

## 3. Discussion

Although antibacterial effects of phytochemicals have been mainly demonstrated to directly affect their viability [21], crude extracts and their isolated compounds could also be promising anti-adhesive and anti-biofilm agents. This strategy would strongly decrease the bacterial colonization by affecting its biofilm formation and adhesion when treating infectious diseases [7,23]. In this study, we have demonstrated that *P. americana* seed-derived ethanol extracts (i) decreased the bacteria cell viability, (ii) reduced the bacterial adhesion to A549 lung epithelial cells probably by inhibiting the mRNA expression levels of genes of virulence factors related to adherence and quorum sensing, (iii) inhibited the biofilm formation, and (iv) were no cytotoxic to mammalian cell lines.

In previous studies, we have found that ethanol extracts from avocado seeds contained several bioactive phytochemicals. Secondary metabolites such as alkaloids, coumarins, tannins, and flavonoids were qualitatively identified by a phytochemical screening [24]. Further characterization analysis also showed the presence of some phenolic compounds such as catechin, as well as neolignan and fatty alcohols/acids as main components [19]. Nevertheless, quantitative differences in phenol compositions and higher phytochemical complexity in SE were found [19]. In this study, we corroborated a similar phytochemical profile and complexity by performing a UHPLC-DAD-QTOF analysis (Appendix A). These works are in accordance with other studies that have identified various classes of phytochemicals mainly polyphenols. However, there is a consensus that these substances, and their quantities, vary with the variety of avocado and may also be influenced by the extraction method performed [15,16]. Some of the lipophilic components present in avocado seed oil as terpenoids and fatty acid derivatives were also extracted [25].

The current study showed that the SE had the highest antibacterial activity against both *S. aureus* and *P. aeruginosa*. Our results are in accordance with previous studies, where ethanol extracts showed similar effects against both types of bacteria [17,26]. The authors also declared that no significant effect was observed in *E. coli*, as confirmed in our study. Some of the secondary metabolites present in SE and MaE, such as catechin, neolignan, and their related compounds, have been widely reported for their antimicrobial effects [27,28,29]. These bioactive molecules have been also suggested to synergize with some of the conventional antibiotics, enhancing their effect and delaying the emergence of drug resistance [30]. Therefore, the antibacterial effect found in SE and MaE might be mainly attributed to previously identified polyphenols, such as catechin and neolignans [19,24].

Besides the direct antibacterial activity, we also investigated the effect on bacterial adhesion to epithelial cells. Both extracts showed promising activities against *P. aeruginosa* and *S. aureus* adhesion to epithelial cells. To the best of our knowledge, this is the first study that reveals avocado seed-derived constituents present in bioactive extracts as anti-adhesive agents when blocked bacteria host-cell attachment. Importantly, the extracts did not affect the viability of A549 lung cells at concentrations 10 times higher than the doses used to inhibit the bacterial attachment Therefore, both extracts could be classified as selective for *S. aureus* and *P. aeruginosa*.

Considering the clinical importance of *P. aeruginosa* in healthcare-associated infections [14,31,32] and the abovementioned results, further studies were performed to investigate how the extracts modify the biofilm development, impair the membrane structure, and alter virulence and adhesion-related factors in *P. aeruginosa*.

An increasing number of studies have revealed the anti-biofilm and anti-adhesion potential of many plant-derived extracts and phytochemicals [3,6,12]. In general, natural anti-biofilm agents can prevent bacterial adhesion, inhibit biofilm maturation, disrupt the biofilm extracellular polymeric substances matrix, and kill microorganisms in mature biofilms [3]. The mechanisms of biofilm formation are regulated by the cell-to-cell communication mechanism termed “quorum sensing” [33]. We demonstrated that the extracts might interrupt the biofilm growth at the initial stage by affecting bacterial communication and adherence mechanisms, and certain virulence factors in *P. aeruginosa*.

It is well-known that *P. aeruginosa* contains a huge library of virulence genes that promote colonization and infection of different tissues [34,35,36,37]. In the present study, ethanol extracts were able to downregulate the *mexT* and *lasA* genes, which belong to the arsenal of pathogenic factors in *P. aeruginosa*. Although *mexT* is mainly involved in multi-drug resistance mechanisms, it has also been reported to indirectly regulate quorum sensing-related phenotypes in *P. aeruginosa* [35,38]. LasA and LasB are extracellular elastases released by this bacterium and implicated in invasion during acute infections [39]. LasA is encoded by the *lasA* gene which also participates in biofilm formation and the quorum sensing network, mainly at the top of the cascade [14,25,40].

Previous studies have highlighted the administration of plant phenolic compounds to control or suppress gene promoters associated with quorum sensing, adherence features, and biofilm development in *P. aeruginosa* [3,10]. This was correlated with the inhibition of the Las system in the quorum sensing network that is controlled by the *lasA* gene. Based on the phytochemical composition from our previous study and our current results, we hypothesize that phenolic compounds present in avocado seed extracts can act through inhibitory mechanisms of the attachment of bacteria and the growth of biofilms. Catechin, neolignan and related compounds, i.e., some of the main phytochemicals identified in *P. americana* seed ethanol extracts, have been reported to prevent the formation of biofilms by altering bacterial attachment, reducing bacterial motility and inhibiting the production of virulence factors regulated by the QS in *P. aeruginosa* [27,29,41].

Gram-negative bacteria such as *P. aeruginosa* contain an external membrane that may act as a barrier for many molecules, which can explain their intrinsic resistance [42]. We demonstrated that the extracts destabilize the membrane integrity of *P. aeruginosa* at concentrations lower than the direct antibacterial activity. The leakage of genetic material after membrane disruption has been proposed as a good indicator for membrane permeability [43]. As it was previously mentioned, phenolic compounds have different antibacterial mechanisms of action. One of those mechanisms is associated with the induction of membrane destabilization and reducing the membrane fluidity [44].

Therefore, taking together all the evidence obtained in *P. aeruginosa*, the antimicrobial, anti-biofilm, and anti-adherence effects of SE and MaE extracts might be attributed not only to different bactericidal mechanisms but also by modulating the expression of virulence factors involved on quorum sensing system, mainly in biofilm growth and adherence properties.

## 4. Materials and Methods

### 4.1. Material and Reagents

Resazurin sodium salt (7-hydroxy-3H-phenoxazin-3-one-10-oxide), L-glutamine, sodium pyruvate, tamoxifen, isoniazid, doxycycline, ciprofloxacin, non-essential amino acid and ultrapure water were purchased from Sigma-Aldrich (St. Louis, MO, USA). Dulbecco’s modified Eagle medium (DMEM), Dulbecco’s phosphate-buffered saline (DPBS), RPMI-1640 medium, and fetal bovine serum (FCS) were from Gibco^®^ (New York, NY, USA). Ethanol ≥ 96% (*v*/*v*) was purchased from VWR (Philadelphia, PA, USA). Mueller-Hinton broth (MHB) was from Difco and Mueller Hinton agar (MHA) and tryptic soy agar (TSA) from Sigma-Aldrich. RNeasy kit was purchased from QIAGEN (Germantown, MD, USA), and RNase-free Dnase I from Ambion and Penicillin-Streptomycin (10,000 U/mL) from Thermo Fisher, respectively (San Jose, CA, USA). GelRed, Random Primers (Catalog number: 48190011) from Invitrogen^®^ and Applied Biosystems™ PowerUp™ SYBR Green Master Mix™ were purchased from ThermoFisher Scientific (San Jose, CA, USA).

### 4.2. Plant Samples Preparation

The selection of plant samples and the preparation procedure were performed following previously standardized and described methodologies [19,24]. Briefly, *P. americana* seeds were obtained from fresh avocado fruits and stored at −20 °C until use. After removing the thin skin, seeds were striped with a grater, using 100 g samples for each raw extract preparation. Ethanol extracts were prepared using Soxhlet (SE) and Maceration/Stirring (MaE) methodologies. Ethanol for analysis was used as solvent obtaining a final volume of 200 mL. Afterwards, ethanol extracts were filtered using a Buchner funnel and vacuum concentrated and dried in an IKA-Werke rotary evaporator RV 3 V model (GmbH and Co.KG, Staufen, Germany) at 45 °C. Then, the residues were dissolved in ultra-pure water by using Vortex and thermostatic bath. The crude extracts were then filtered (0.2 µm, ThermoFisher Scientific, USA) and aliquots were stored at −20 °C until use (stock solutions: 100 mg/mL). Phytochemical composition by UHPLC-DAD-QTOF analyses for each ethanol extract was determined based on main peak comparison (retention time, MM and fractioning pattern with the previously analyzed and characterized extracts [19]. The accurate mass measurements were carried out using a Xevo G2-XS QTOF spectrometer (Waters, Milford, MA, USA) coupled with an ACQUITY LC system equipped with MassLynx software version 4.1. (Waters Inc., Milford, MA, USA) Data were recorded using MS^E^ in negative ionization mode, and ramp collision energy from 10 to 30 V was applied to obtain additional structural information. The mobile phase consisted of H_2_O + 0.1% FA (A) and can +0.1% FA (B).

### 4.3. Cell Lines, Bacterial Strains, and Culture Conditions

(A549 human lung cells (ATCC CCL-185™) and MRC-5 SV2 human fetal lung fibroblasts from ATCC (American Type Tissue Culture Collection, Manassas, VA, USA) were maintained in DMEM supplemented with 10% inactivated fetal bovine serum (FBS), L-glutamine (1%), sodium pyruvate (1%), non-essential amino acid (1%), and antibiotics (P/S). Cell cultures were kept in T75 flasks at 37 °C in a humidified atmosphere with 5% CO_2_.

All bacteria were obtained from different culture collections: *Pseudomonas aeruginosa* ATCC 9027, *Pseudomonas aeruginosa* ATCC 27853, *Escherichia coli* ATCC 8739, *Staphylococcus aureus* ATCC 6538, and *Streptococcus pneumoniae* NTCC 7466. *P. aeruginosa*, *E. coli,* and *S. aureus* strains were cultured in Mueller-Hinton broth (MHB, Difco), and *S. pneumoniae* in Mueller Hinton agar supplemented 5% lysed horse blood (MHA, Sigma-Aldrich, St. Louis, MO, USA). In the case of adherence assays *P. aeruginosa*, *E. coli* and *S. aureus* were grown on Mueller-Hinton agar and *S. pneumoniae* on TSA with 5% defibrinated sheep blood.

### 4.4. Cytotoxicity Assessment

Briefly, A549 and MRC-5 SV2 cells (5 × 10^5^ cells/mL) were seeded in sterile 96-well plates, containing 10 µL of the test extracts at different concentrations ranging from 32 to 1024 µg/mL, and incubated at 37 °C, 5% CO_2_ for 24 h. Tamoxifen was included as a reference compound. The cell viability was estimated fluorometrically 4 h after the addition of 50 µL/well of 0.01% resazurin solution using a microplate reader (TECAN GENios, Männedorf, Switzerland) at *λ*_ex_ 550 nm, *λ*_em_ 590 nm [45]. The results were expressed as per cent reductions in viability compared to control (untreated) wells and median cytotoxic concentration (CC_50_) was determined.

### 4.5. Antimicrobial Assessment

#### 4.5.1. Antimicrobial Screening

##### Microdilution Method

Assays were performed in sterile flat-bottomed 96-well microtiter plates. Each well contained 10 µL of the extract dilutions together with 190 µL of bacteria inoculum (5 × 10^3^ CFU/mL). Bacterial growth is compared to untreated-control wells (100% cell growth) and medium-control wells (0% cell growth). Plates were incubated for 16 h (*E. coli*, *P. aeruginosa*, *S. aureus,* and *S. pneumoniae*). After incubation, antibacterial activity was assessed by the resazurin test. Then, 20 µL of a 0.01% (wt/vol) resazurin solution was added to each well and plates were incubated for 15 min (*S. aureus*), 30 min (*E. coli*), or 45 min (*P. aeruginosa* and *S. pneumoniae*) to allow resazurin reduction to take place. Fluorescence was read at *λ*_ex_ 550 nm and *λ*_em_ 590 nm. The results are expressed as % reduction in bacterial growth compared to control wells and IC_50_ (50% inhibitory concentration) values are determined. Doxycycline (for *E. coli*, *S. aureus,* and *S. pneumoniae*) and ciprofloxacin (for *P. aeruginosa*) were used as reference drugs. For pure compounds (natural isolates) antibacterial activity is explored between 2-64 µg/mL [46], nevertheless. For avocado extracts, there is consent that IC_50_ values below 500 µg/mL extracts can be classified as active [26,47]. Therefore, a concentration range from 1024 until 32 µg/mL (1024, 512, 254, 128, 64, and 32) was tested and the limit for activity was set at 500 µg/mL.

##### Bacterial Adherence Assay

For the bacterial adherence assays, extracts were tested at concentrations lower than their IC_50_ values obtained in the antibacterial test. For *P. aeruginosa* and *S. aureus* a 64 µg/mL concentration was used, while for *S. pneumoniae* and *E. coli* a 128 µg/mL and 512 µg/mL were used, respectively. Briefly, A549 cells (2 × 10^5^ cells/mL) were incubated for 24 h in 24-well plates (5% CO_2_, 37 °C). Afterward, medium was discarded and bacteria (*E. coli* ATCC 8739, *P. aeruginosa* ATCC 9027, *S. aureus* ATCC 6538, and *S. pneumoniae* NTCC 7466) grown at mid log phase (2 × 10^6^ CFU/mL) on medium diluted were added to the 24-well plate at a multiple of infection (MOI) of 10. Then, plant extracts were immediately added to the co-culture. After one hour of incubation (5% CO_2_, 37 °C), the co-culture was washed three times with DPBS to remove the unbound bacteria. Finally, the adhered bacteria were detached using 0.1% Triton X-100 and the viability was analyzed by using the standard viable plate count method [48].

#### 4.5.2. Biofilm Formation Assay

A biofilm formation assay was performed following the standardized protocol “Crystal Violet Assay Bacteria Biofilms” in 96-well plates, according to Toté et al. (2009) [49]. *P. aeruginosa* ATCC 9027 was grown overnight in MHB at 37 °C. Then, bacterial suspensions were transferred to flat 96-well microtiter plates and ethanol extracts were added. Bacteria with medium only were considered as negative control. The plates were incubated during 24 h at 37 °C to allow biofilm formation. After incubation, the medium was discarded and wells were washed with phosphate-buffered saline. Adherent cells were then fixed with 99% ethanol solution for 15 min. Ethanol was removed and wells were air dried for 2 min. The attached cells were stained with 0.10% (*w*/*v* in 12% ethanol) crystal violet for biofilm formation. Wells were then rinsed with DPBS and blotted dry. For quantification of biofilm growth, cells associated with crystal violet were solubilized with absolute ethanol, and absorbance was measured at 570 nm using a microplate reader (TECAN GENios, Männedorf, Switzerland).

#### 4.5.3. Measurement of Release of Nucleic Acids

The measurement of the release of nucleic acids into the supernatant from *P. aeruginosa* as an indirect method to assess the bacterial membrane impairment was performed following the methodology proposed by Liu et al. 2020 [50]. *P. aeruginosa* suspensions (10^6^ CFU/mL) were incubated at 37 °C and shaken at 150 rpm in the presence of plant extracts (128 µg/mL) during 6 h and 24 h at 37 °C. Then, the suspensions were collected and centrifuged at 4000 rpm for 10 min. The optical density (OD) was measured at 260 nm by using a Qubit^®^ 4 Fluorometer (Thermo Fisher Scientific, San Jose, CA, USA). The experiment was performed in triplicate.

### 4.6. Real Time PCR and Gel Electrophoresis Analysis

*Pseudomonas aeruginosa* ATCC 27853, a strain that contain distinctive expression profiles of the virulence genes was chosen for this study [51]. Both treated (with SE and MaE extracts at 64 µg/mL) and untreated (control) bacteria were lysed and total RNA was isolated using an RNeasy kit (QIAGEN) in accordance with the manufacturer’s instructions. The purified RNA was quantified by spectrophotometry BioDrop µLite+ (Biochrom Ltd., Cambridge, UK) at 260 nm. Total RNA (0.7 µg) was then treated with the DNase I, RNase-free, and reverse transcribed using Random Primers. The primers used (forward and reverse) are shown in Table 3.

Real-time RT-PCR was performed using Applied Biosystems™ PowerUp™ SYBR Green Master Mix™ (ThermoFisher Scientific, CA, USA) and a PikoReal 96 Real-Time PCR System (ThermoFisher Scientific, San Jose, CA, USA). Amplification was carried out with the following cycle profile: one cycle of initial denaturation at 95 °C for 2 min, followed by 40 cycles of 95 °C for 15 s, 56 °C for 25 s, and 72 °C for 30 s [22]. The threshold cycle values (Ct) were automatically determined for each reaction. The relative mRNA gene expressions levels were calculated through the value obtained between ΔCt values of treated and untreated bacteria (Control) (*n* = 4). The *creB* gene was used as an internal control.

PCR products were also analyzed by agarose gel electrophoresis (1%) stained with GelRed using a Tris-borate-EDTA buffer. Gels were visualized by OmniDOC Gel Documentation System (Cleaver Scientific, Rugby, UK).

### 4.7. Statistical Analysis

The data were analyzed by using the GraphPad Prism 7 (Windows, V. 7.04, 2017, San Diego, CA, USA) software package. The results were statistically analyzed and expressed as the arithmetic means ± standard deviation (SD). The statistic differences on all experiments were determined by a one-way ANOVA test followed by the Tukey’s test. Differences at *p* ≤ 0.05 were accepted as significant. Two independent assays were performed by duplicate in each experiment at minimum.

## 5. Conclusions

Our findings indicate that the ethanol extracts from *P. americana* seeds, which are rich in phenolic substances, can inhibit bacterial growth by directly affecting the cell viability and by interrupting the bacterial attachment to lung epithelial cells, without triggering cytotoxic effects on mammalian cell lines. The anti-biofilm and anti-adhesion activities of the ethanol extracts on *P. aeruginosa* might be associated with the modulation of the quorum sensing system by down-regulation of the virulence factors such as *mexT* and *lasA* genes. Bioproducts from avocado seeds may be considered as promising anti-biofilm and anti-adhesive agents.

## Figures and Tables

**Figure 1 molecules-27-05009-f001:**
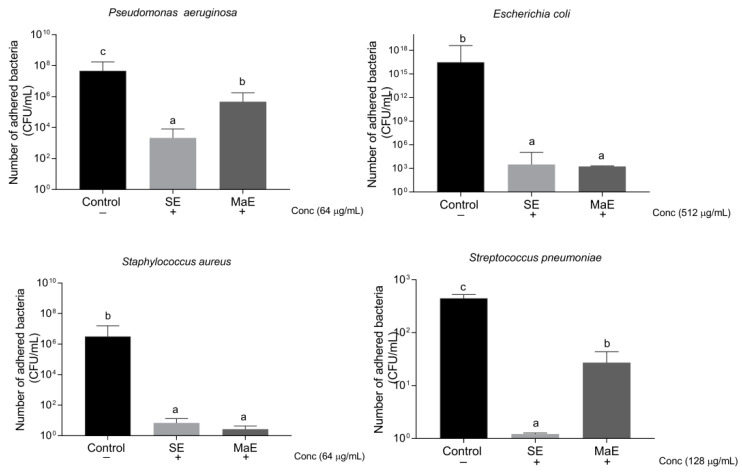
Effect of *P. americana* seed ethanol extracts on bacterial adhesion to A549 lung epithelial cells. A549 cells (2 × 10^5^ cells/mL) were incubated for 24 h (5 % CO_2_, 37 °C). Supernatant was discarded and bacteria (*E. coli* ATCC 8739, *P. aeruginosa* ATCC 9027, *S. aureus* ATCC 6538, and *S. pneumoniae* NTCC 7466) grown at mid log phase (2 × 10^6^ CFU/mL) on medium diluted with test extracts were co-cultured at a multiple of infection (MOI) of 10. After one hour incubation the co-culture was washed with DPBS to remove the unbound bacteria. Adhered bacteria were detached using 0.1% Triton X-100. Adhesion was analyzed by using the standard viable plate count method. All values are expressed as the arithmetic mean ±SD. Means without the same letter are significantly different at the 5% level according to a one-way ANOVA test followed by a Tukey test (*n* = 3).

**Figure 2 molecules-27-05009-f002:**
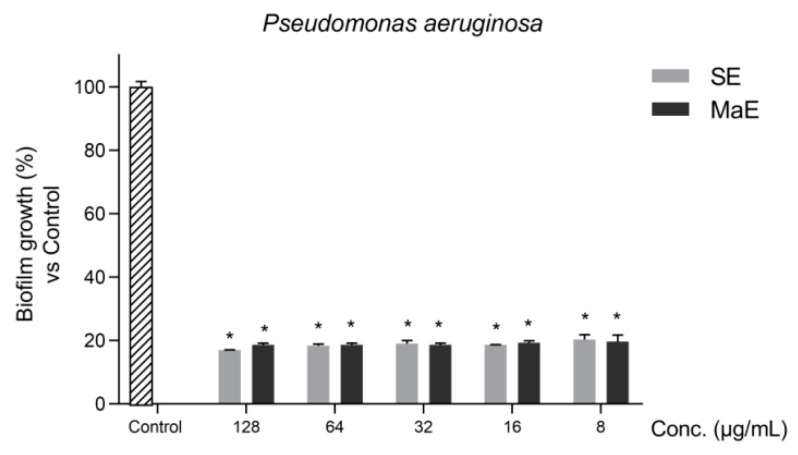
Inhibition of *P. aeruginosa* ATCC 9027 biofilm growth by *P. americana* seed ethanol extracts. SE: ethanol extract obtained by Soxhlet (SE); MaE: ethanol extract obtained by maceration. Biofilms were stained with 0.10 % crystal violet and quantified by absorbance at 570 nm. Bacteria with medium only were considered as negative control. All values are expressed as the arithmetic mean ± SD. (*) Significant differences at the 5 % level according to one-way ANOVA test followed by the Tukey’s test (*n* = 4).

**Figure 3 molecules-27-05009-f003:**
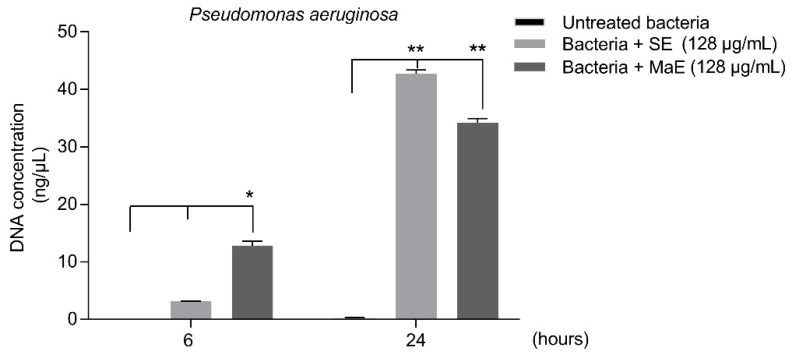
Effect of *P. americana* seed ethanol extracts on *Pseudomonas aeruginosa* ATCC 9027 membrane integrity. SE: ethanol extract obtained by Soxhlet; MaE: ethanol extract obtained by maceration. *P. aeruginosa* membrane impairment was evaluated by measuring the release of nucleic acids. *P. aeruginosa* suspensions were incubated at 37 °C in the presence of plant extracts (128 µg/mL) during a 4h period. After centrifugation, supernatants were collected at 6 and 24 h and the optical density (OD) was recorded at 260 nm. All values are expressed as the arithmetic mean ± SD. (**) and (*) Significant differences at 1% and 5 % levels according to one-way ANOVA test followed by the Tukey’s test (*n* = 4).

**Figure 4 molecules-27-05009-f004:**
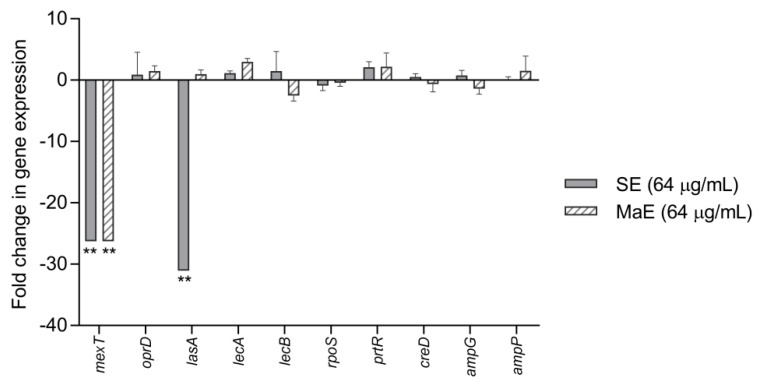
Effect of *P. americana* seed ethanol extracts on expression of virulence-related genes in *Pseudomonas aeruginosa* ATCC 27853. SE: ethanol extract obtained by Soxhlet; MaE: ethanol extract obtained by maceration. The mRNA expression levels of genes were calculated through the value obtained between the ΔCt values of treated and untreated bacteria (Control). *creB* gene was used as internal control. All results are expressed as mean ± SD. (**) Significant differences at the 1 % level according to one-way ANOVA test followed by the Tukey’s test (*n* = 4).

**Figure 5 molecules-27-05009-f005:**
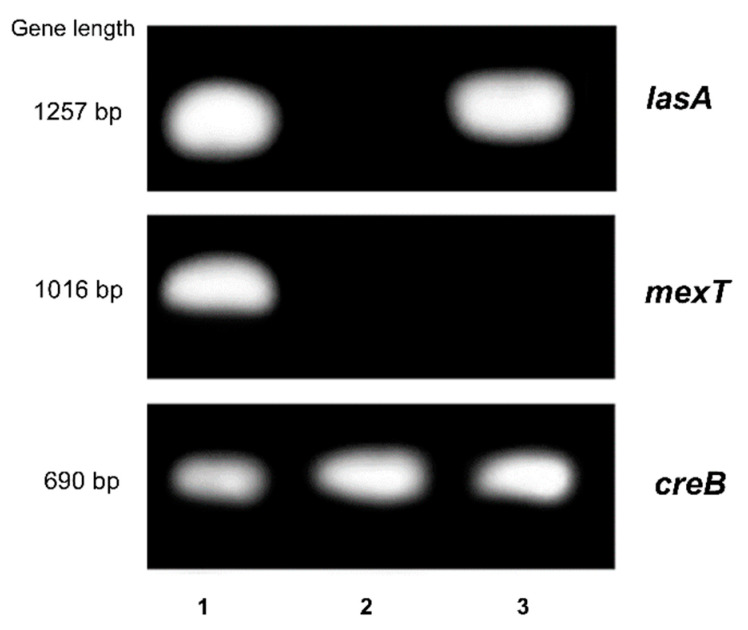
Electrophoresis analysis of *lasA* and *mexT* genes in *Pseudomonas aeruginosa* ATCC 27853. SE: ethanol extract obtained by Soxhlet; MaE: ethanol extract obtained by maceration. PCR products were analyzed by agarose gel electrophoresis (1%) stained with GelRed and using a Tris-borate-EDTA buffer. Gels images of *lasA* and *mexT* genes were visualized. Lanes: (1) Control: untreated bacteria, (2) SE treatment, (3) MaE treatment. *creB* gene was used as internal control.

**Table 1 molecules-27-05009-t001:** Antibacterial effects of *P. americana* seed extracts.

Bacteria	IC_50_			
	SE (µg/mL)	MaE (µg/mL)	Doxycycline (µM)	Ciprofloxacin (µM)
*E. coli* ATCC 8739	˃1024	˃1024	0.50 ± 0.28	-
*P. aeruginosa* ATCC 9027	87.0 ± 4.4 *	187.4 ± 9.4 *	-	1.05 ± 0.50
*S. aureus* ATCC 6538	144.2 ± 5.7 *	159.2 ± 7.9	0.04 ± 0.95	-
*S. pneumoniae* NTCC 7466	227.6 ± 11.4	532.2 ± 26.6	3.56 ± 0.84	-

SE: ethanol extract obtained by Soxhlet; MaE: ethanol extract obtained by maceration. Results are expressed by means ± SD and represent two independent experiments performed by duplicate each one. (*) significant differences at the 5% level to according to one-way ANOVA test followed by a Tukey’s test. Doxycycline and ciprofloxacin were used as reference control drugs.

**Table 2 molecules-27-05009-t002:** In vitro cytotoxicity of *P. americana* seed extracts.

Cell Lines	CC_50_		
	SE (µg/mL)	MaE (µg/mL)	Tamoxifen (µM)
MRC-5	636.9 ± 31.8	786.3 ± 39.3	8.7 ± 0.4
A549	770.0 ± 38.5	774.3 ± 38.7	6.1 ± 0.3

SE: ethanol extract obtained by Soxhlet; MaE: ethanol extract obtained by maceration. The effect of the extracts on cell viability was determined by the resazurin reduction assay. All values are expressed as the arithmetic mean ± SD of two independent experiments. Tamoxifen is a reference control drug for cytotoxicity.

**Table 3 molecules-27-05009-t003:** Gene and primers used for quantitative reverse-transcriptase polymerase chain reaction (sequence 5′-3′).

Gene	Forward Primer	Reverse Primer
*mexT*	*ACCTCATGGGTTGTGACTGTATCC*	*TAGGATCACTGACAGGCATAGCCA*
*oprD*	*TTTCCGCAGGTAGCACTCAGT*	*CTTCGCTTCGGCCTGATC*
*lasA*	*TTCTGTGATCGATTCGGCTCGGTT*	*ACCCGGGAAGACAACTATCAGCTT*
*lecA*	*CACCATTGTGTTTCCTGGCGTTCA*	*AGAAGGCAACGTCGACTCGTTGAT*
*lecB*	*AGACAGCGTAACAATCGAACGAGC*	*AGGACGCATCGTTCAGCCAATCTA*
*rpoS*	*CGGCGAGTTGGTCATCATCAAACA*	*ATCGATTGCCCTACCTTGACCTGT*
*prtR*	*TCCCTGCACCCATGTGAAATCTCT*	*ATCGGCAATCTACAGACCGATGGA*
*creD*	*CGCCATCGCCCTACTCAT*	*GGCGATCGCGGATCAG*
*ampG*	*GGCCTGACCTATGCGTTGTT*	*CGACGGGTTGACGTGGAT*
*ampP*	*ACCGCCGCCTGGTCAT*	*CAGGCCGATGGGAATGC*
*creB*	*GCATATCCTGATCGTCGAAGATG*	*GGCCTGCAGGGCGTAGA*

## Data Availability

Not applicable.

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
