# Peer review of "Inhibition of Bacterial Adhesion and Biofilm Formation by Seed-Derived Ethanol Extracts from *Persea americana* Mill"

_molecules, 2022, doi:10.3390/molecules27155009_

Round 1
Reviewer 1 Report
In the manuscript of “Inhibition of bacterial adhesion and biofilm formation by seed derived ethanol extracts from Persea americana Mill”, the interesting data have showed that avocado seed extracts were anti adhesive and anti-biofilm agents. The study is meaningful. However, there are some issues in the manuscript:
1. As a journal of Molecules, a certain molecule might be appropriate, at least, the weight of molecules should be fixed. What are the molecules that performed the function? As the solution is ethanol, both hydrophilic and hydrophobic substances are possible to be extracted. The extracts (SE and MaE) should be filtered by different sizes of filter. Then, bio-macro-molecule could be removed.
2. Avocado seeds oil is commercialized, the extracts in this study are exist or not in the oil. Whether the extracts could be obtained from the avocado seeds meal?
3. For the control, the stock solutions were prepared in ethanol, so, the negative control should be set two: untreated and ethanol. Ethanol might affect the results.
4. Figure 3, miss the ANNVA test for the 6 h.
5. Figure 5, for the electrophoresis gel, a marker should be showed in the gel.
6. The writing style is not homogenous.
Line 31, 35: a space between number and units. Check all the text.
Line 116: minutes : min. 167 hours: h,
Line 399, CFU? Or CFR/mL?
7. Line 413, suggestion: list a table for the primers.
Author Response
"Please see the attachment"

Reviewer 2 Report
Please see the attachments.

Author Response
"Please see the attachment"

Round 2
Reviewer 1 Report
For the revised manuscript, the authors have addressed all the questions; therefore, I recommend the manuscript for publication.